# TGCA-PVT: Topic-Guided Context-Aware Pyramid Vision Transformer for Sticker Emotion Recognition

## ABSTRACT

Online chatting has become an essential aspect of our daily interactions, with stickers emerging as a prevalent tool for conveying emotions more vividly than plain text. While conventional image emotion recognition focuses on global features, sticker emotion recognition necessitates incorporating both global and local features, along with additional modalities like text. To address this, we introduce a topic ID-guided transformer method to facilitate a more nuanced analysis of the stickers. Considering that each sticker will have a topic, and stickers with the same topic will have the same object, we introduce a topic ID and regard the stickers with the same topic ID as topic context. Our approach encompasses a novel topic-guided context-aware module and a topic-guided attention mechanism, enabling the extraction of comprehensive topic context features from stickers sharing the same topic ID, significantly enhancing emotion recognition accuracy. Moreover, we integrate a frequency linear attention module to leverage frequency domain information to better capture the object information of the stickers and a locally enhanced re-attention mechanism for improved local feature extraction. Extensive experiments and ablation studies on the large-scale sticker emotion dataset SER30k validate the efficacy of our method. Experimental results show that our proposed method obtains the best accuracy on both single-modal and multi-modal sticker emotion recognition.

## CCS CONCEPTS

• **Computing methodologies** → **Artificial intelligence**; • **Information systems** → **Information systems applications**.

## KEYWORDS

sticker emotion analysis, multimodal learning, sentiment analysis

## 1 INTRODUCTION

With the popularity of the internet and information techniques, online chatting has become an indispensable part of our daily lives. In the process of online chat, in addition to using pure text messages for communication, users often use stickers, one kind of image with abundant information, for better expression. The sticker, as an effective carrier of pictures and text [13], often plays the role of a picture worth a thousand words in the chat process, which can effectively reflect the emotions of users [28]. Therefore, recognizing

*ACM MM, 2024, Melbourne, Australia*
© 2024 Copyright held by the owner/author(s). Publication rights licensed to ACM.
ACM ISBN 978-x-xxxx-xxxx-x/YY/MM
https://doi.org/10.1145/nnnnnnn.nnnnnnn

the emotion of the sticker used in chatting can help us understand the emotion of the conversation.

In recent years, image-based emotion recognition which is one of the most important parts of visual affective analysis [18], especially the realistic images [33] and facial expressions [24]. Complicated visual elements are contained in the image including both low-level and high-level, local and global. With the availability of large-scale data sets and the continuous development of deep learning methods, there has been a lot of research on pre-training networks on large-scale data sets and migrating to image emotion recognition tasks [22, 36]. These methods have achieved encouraging results in image-based emotion recognition. However, compared to normal images, stickers can be comprehensive representations of various visual elements such as cartoon characters, facial expressions, and textual illustrations [2]. Due to the complexity of information and lack of sufficient data, few studies have targeted specific emoji recognition. Ref [17] proposed the first large-scale Sticker Emotion Recognition dataset called SER30K, which provided the basis and convenience for sticker emotion recognition.

Recognizing the emotion of stickers is more challenging than realistic image sentiment recognition since both the local and global information is the same important. Despite being frequent users of stickers, there are differences in how the sender and the recipient interpret the emotions of the stickers [29]. Stickers are usually grouped by theme, and stickers under the same theme will have multiple different emotional labels. On the one hand, the theme can help with finding the subject of the stickers for the global feature extraction, which can set the emotional tone. On the other hand, the differences between the emotions of the stickers from the same theme will be depicted in the details, which are also the local features. Therefore, to fully explore the global features and local features is crucial for sticker emotion recognition.

Against this background, we proposed a novel topic-guided context-aware method to capture both global and local features of stickers. Firstly, we assign each sticker a topic ID. Stickers that come from the same original sticker or with the same theme will share the same topic ID as Fig. 1. We further regard the feature from the stickers with the same topic ID as the context feature of the stickers. As we can see in Fig. 1, conducting context-aware on the sticker and its transformed version can achieve local detail enhancements. For the stickers with the same theme, context-aware can help the model focus on the subject of the sticker and better grasp the global feature. Specifically, we design a novel **t**opic-**G**uided **c**ontext-**a**ware module (TGCA-Module) and introduce it into a pre-trained pyramid vision transformer (**PVT**) model [31], which is called **TGCA-PVT** to better improve the performance on sticker emotion recognition. We also design a **l**ocally **e**nhanced **re**-**a**ttention (LERA) to enhance local details and a **t**opic **g**uided **a**ttention (TG-Attention) to enhance the global features according to topic ID. We conduct extensive experiments on the large-scale sticker emotion recognition dataset

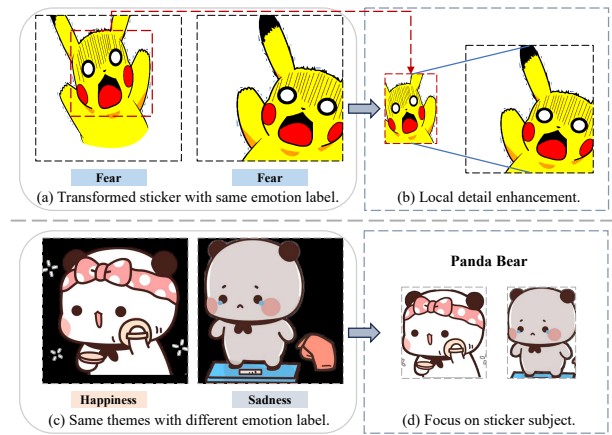

(a) Transformed sticker with same emotion label.

(b) Local detail enhancement.

(c) Same themes with different emotion label.

(d) Focus on sticker subject.

**Figure 1: Examples of stickers sharing the same topic ID. (a) Image transformations, like altering the image, maintain the topic ID and emotional labels. Conducting topic context-awareness on these can enhance local details, as seen in (b). (c) Stickers under the same topic ID but with varying emotional labels. Conducting topic-guided context-awareness on these can aid in better understanding the image subject and extracting effective features, as shown in (d).**

SER30K [17] and image emotion recognition dataset FI [40] to evaluate our proposed method comprehensively. The main contributions of this work can be summarized as follows:

- Based on the subject information of emoticons and the data enhancement methods commonly used in the image processing field, we introduce the concept of Topic ID and propose a TGCA-Module and a TG-Attention based on Topic ID to mine the subject information shared by emoticons with the same subject, as well as the local enhancement features brought by the image transformation.
- We design a novel frequency linear attention module (FLA-Module) to better capture the frequency domain information for sticker object feature extraction and a novel LERA-Module to the ability of the proposed model to extract local details of stickers for better emotion recognition.
- Extensive experiments and ablation studies are conducted on the public large-scale sticker emotion recognition dataset SER30K to verify the effectiveness of our proposed method.

## 2 RELATED WORK

### 2.1 Image-based emotion recognition.

Affective image recognition is now widely conducted to analyze how humans feel about what they see in the computer vision field [46]. One of the biggest challenges is to fill the affective gap between bitmap and abstract emotion [12]. Previous methods generally use CNNs to capture the global features of the image and then analyze the emotion contained [43]. Chen *et al.* [3] and Peng *et al.* [20] used deep CNN models to classify the image emotion and demonstrated that using deep CNN models is superior to previous handcrafted techniques and machine learning-based methods (such as SVM).

Furthermore, Zhu *et al.* [49] proposed a multi-task learning framework that leverages CNN to extract different levels of features and RNN to integrate the learned features. Ankita *et al.* [21] also combined RNN and CNN better to integrate the multi-level visual attributes for sentiment classification. Furthermore, the emotional cues are thought to be contained in the regional features or details of the image, therefore, some recent studies focus on fully exploiting the visual emotion with regional features or shallow visual details. Yang *et al.* [38] first used an off-the-shelf objectness tool to capture features of the object regions and utilized CNN to compute the corresponding sentiment scores of each region, then aggregated these features to obtain the prediction of image emotion. Rao *et al.* [22] proposed the MldrNet to consider global and local image views and capture image semantics, aesthetics, and low-level visual features for emotion recognition. Considering the success of the feature pyramid network, Rao *et al.* [23] proposed a region-based deep CNN model to capture multi-level features for visual sentiment recognition. Although these methods can effectively capture the emotional features of images, they are not completely suitable for sticker emotion recognition because of the specificity of stickers. Local features, object relationships, theme information, and even texts are all the emotion-related information of stickers, and that's what we consider in our work.

### 2.2 Sticker and Emotion.

Stickers, which have more expressive information including diverse animations, multiple objects, and texts than emoticons and emojis, are considered one of the fundamental features of instant messaging [25]. As multi-modal information [27], stickers can help people fill in important information lost during online chats, such as gestures and facial expressions, in face-to-face conversations [7]. Similar to emoticons and emoji, stickers can enhance human interaction in online chatting, and also improve the expression of emotion for people [5]. The sticker theme represents the character image adopted within a sticker package, which includes TV series, cartoons, anime, etc. Users could utilize stickers in cartoon themes with detailed illustrations to express their emotions, such as intimacy [30]. Because of their expressive power, stickers have more advantages for emotional intensity, positivity, and intimacy [16], but they also have drawbacks, that as emotional misinterpretation [2]. Since the sticker must be sent as a separate message, emotional misinterpretation of stickers often occurs more frequently than emoticons and emojis. Thus, it can be seen that achieving accurate emotion recognition of stickers is a very challenging task, and it is also more meaningful for machines to understand human emotions.

## 3 METHODOLOGY

We design our TGCA-PVT with the pre-trained PVT model and Bert model as the backbone network. For the characteristics of emotion recognition of emoticons, several modules are introduced to further improve the accuracy of emotion recognition. The whole framework of our proposed TGCA-PVT is shown in Fig. 2.

### 3.1 Backbone Network

Considering the great performance of the PVT models in capturing the global relation information of image [31], we adopt a pre-trained

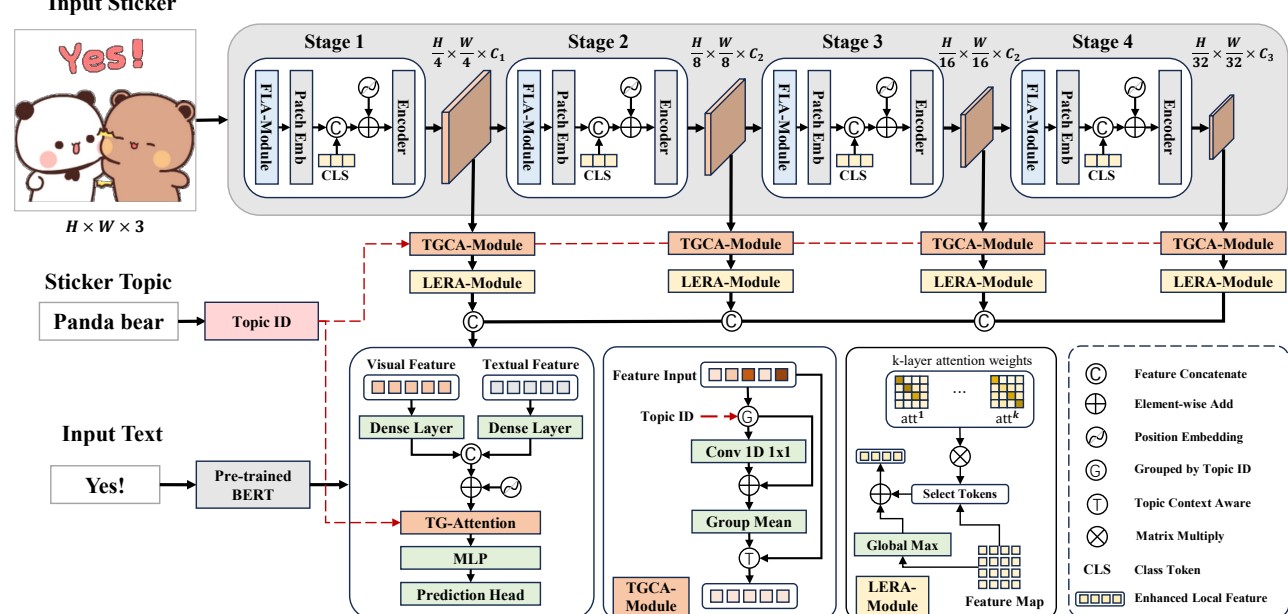

**Figure 2: Overview of our proposed TGCA-PVT model. The PVT and the BERT are used for the visual encoder and the textual encoder respectively. A FLA-Module is designed to capture the frequency domain information. We then design the TGCA-Module to capture the context features from the input with the same topic ID and the LERA-Module to enhance the local feature of stickers. The TG-Attention is designed to enhance the features with topic-guided features for better emotion recognition.**

PVT model, which contains four stages, as the backbone visual encoder. Specifically, given a sticker input $X \in \mathbb{R}^{3 \times H \times W}$, where $H$ and $W$ represent the height and width of the sticker, PVT will first use a Conv2d layer to project it and then flatten it as a sequence of the patch features $X_{patch} \in \mathbb{R}^{N \times C}$, where $N = \frac{HW}{P^2}$ and $P$ is the patch size, this process is also called the Patch Embed. During each stage, we introduce the FLA-Module, which will be illustrated in detail in Section 3.2, to enhance the frequency features of the stickers before feeding them to the Patch Embed. Then we concatenate an additional CLS token $X_{CLS} \in \mathbb{R}^{1 \times C}$ ahead of the patch features to capture the global and local patches better. What's more, we also add the position embedding $X_{pos}$ to deal with the position-agnostic problem of the input patch tokens before feeding it to the encoder of the PVT model. Therefore, the input of the encoder in each stage can be illustrated as Eq. 1.

$$X_{in}^l = Concat(X_{CLS}^l, X_{patch}^l) + X_{pos}^l, X_{in}^l \in \mathbb{R}^{(N+1) \times C} \quad (1)$$

where $X_{in}^l$ represents the input of encoder in the $l$-th stage, and $Concat$ represents the concatenation operation.

The encoder of the PVT model uses spatial-reduction attention (SRA) to replace multi-head attention (MHA) in the first three stages. Similar to MHA, two linear projections are used to obtain the $X_{in}^l$ into the query, key, and value embeddings as Eq. 2 to Eq. 5.

$$Q^l = W_q^l X_{in}^l + b_q^l, \quad (2)$$

$$KV^l = W_{kv}^l X_{in}^l + b_{kv}^l, \quad (3)$$

$$KV'^l = Reshape(KV^l), \quad (4)$$

$$K^l = KV'^l[0], V^l = KV'^l[1], \quad (5)$$

where $Q^l \in \mathbb{R}^{(N^l+1) \times C^l}$ is the query embedding, and the key embedding and the value embedding are obtained by a linear projection so $KV^l \in \mathbb{R}^{(N^l+1) \times (C^l \times 2)}$. Then the $KV^l$ is reshaped to $KV'^l \in \mathbb{R}^{2 \times (N^l+1) \times C^l}$, and final separate it to $K^l, V^l \in \mathbb{R}^{(N^l+1) \times C^l}$. $C^l$ represents the hidden dimension of the $l$-th stage.

The spatial-reduction calculation is shown as Eq. 6:

$$SR(x) = LN(Reshape(X', S_l)W^S), \quad (6)$$

where $X'$ is the spatial feature that obtained by remove the $X_{CLS}$ from $X_{in}$, LN represents the layer normalization, $S_l$ represents the reduction ratio $l$-th stage, and $W^s$ is a learnable linear transformation. The whole process of SRA is shown in Eq. 7 and Eq. 8

$$SRA(Q, K, V) = Concat(head_1, ..., head_{N_l})W^O, \quad (7)$$

$$head_i = Attention(QW_i^Q, SR(K)W_i^K, SR(V)W_i^V), \quad (8)$$

where $head_i$ represents the $i$-th self-attention head, $N_l$ is the head numbers in the $l$-th stage, and $W^O \in \mathbb{R}^{C^l \times C^l}$, $W_i^Q, W_i^K, W_i^V \in \mathbb{R}^{C^l \times d_{head}}$, where $d_{head}$ is the the dimension of each attention head. The CLS token is then projected by a linear transformation $W_{global}$ to better capture the global features $x_g^l$ and then concatenate it with $X$ after SR. The attention head operation is shown as Eq. 9:

$$Attention(Q, K, V) = softmax(\frac{QK^T}{\sqrt{d_{head}}})V \quad (9)$$

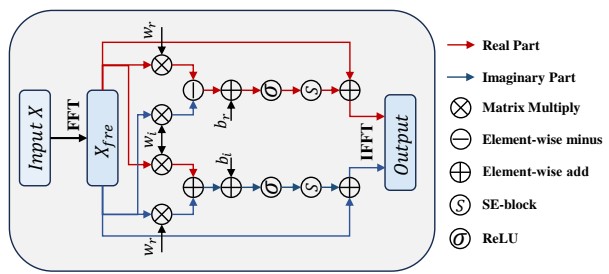

**Figure 3: Framework of the proposed FLA-module.**

Finally, the output feature $X_{out}^l$ of the $l$-th stage is fed to TCA-Module and the next stage. If the sticker input has textual information, we adopted a pre-trained BERT model to obtain the contextualized word representations $X_t$.

### 3.2 FLA-Module

Inspired by Frequency MLP [39], we design a **F**requency **L**inear **A**ttention module (FLA-module) capture the frequency domain information for the object feature enhancement, which can help distinct the object from the background [47]. The proposed FLA-Module is shown in Fig. 3. Given the input $X$, we first use a faster Fast Fourier Transform (FFT) to get the frequency components $X_{fre} = x_r + jx_i$, which is a complex number feature. According to the multiplication of complex numbers, we introduce learnable weights $W = W_r + jW_i$ and $B = B_r + jB_i$ to conduct frequency linear computation as Eq. 10.

$$h_r + jh_i = \sigma(x_r W_r - x_i w_i + B_r) + j\sigma(x_r W_i + x_i w_r + B_i) \quad (10)$$

Then we introduce squeeze and excitation block (SE block) [14] to recalibrate channel-wise feature responses as Eq. 11.

$$h_r' + jh_i' = SE(h_r) + jSE(h_i), \quad (11)$$

where SE represents the SE block. Furthermore, we add the original frequency information and then stack them to obtain a complex number as the output features. Finally, we use the inverse fast Fourier transform (IFFT) to get the frequency-enhanced features $y$ of the input $X$. The whole process is shown as Eq. 12

$$y = IFFT(y_r + jy_i) = IFFT((x_r + h_r') + j(x_i h_i')) \quad (12)$$

### 3.3 TGCA-Module

To better capture the theme information, we first assign each sticker a topic ID as illustrated in Section 1. Given a batch of stickers input $X_{out}^l$, we will first group them by topic id and then a Conv1d layer with a kernel size of 1 is used to capture the channel correlation of the same group of features. A skip connection is used to ensure specificity between features in the same group. The whole process operation is shown as Eq. 14.

$$X_c^l = X_{out}^l + Conv1d(X_{out}^l), \quad (13)$$

Then for each group, we adopt the average of all the stickers as the group context feature, each sticker will obtain the corresponding context feature and we stack them as the context-aware features. Then we also utilize the $W_{global}$ to obtain the global context $x_{cg}^l$.

With the two global features from the CLS token in the encoder and from the context-aware features, we further design a Topic Context Aware layer to fuse them inspired by the attention mechanism. The operation of the Topic Context Aware layer is shown in Eq. **??**.

$$X_g'^l = X_g^l + Softmax(X_g^l(X_{cg}^l)^T)X_{cg}^l, \quad (14)$$

In this way, the global feature can effectively learn the context features of the stickers with the same topic ID. Then we stack the global feature of each stage as $X_{global}$ to represent the global representation of the sticker.

### 3.4 LERA-Module

To better capture the relationship between the region information and the sticker emotion, we proposed a local enhanced re-attention module (LERA-Module) inspired by the Ref. [17]. Since the different local information like the expressions and the poses has different scales, LERA is utilized in each stage of the PVT encoder to exploit multi-scale features. In the attention mechanism of each encoder, we can also obtain the attention weight distribution among patches as Eq. 16.

$$a_l = Softmax(\frac{QK^T}{\sqrt{d_{head}}}), \quad (15)$$

where $a_l$ is the corresponding attention weights in the $l$-th stages. Considering that each encoder has $K_l$ attention layers, we regard the attention weights in each attention layer as Eq. 16.

$$a_l^n = [a_l^{n1}, a_l^{n2}, ..., a_l^{n3}], i \in 1, 2, ..., K_l, \quad (16)$$

where $a_l^n$ is the $i$-th attention head in the $l$-th stage. Then by multiplying the attention weight of each head at the same stage, we can obtain the final attention weights as Eq. 17.

$$a_l^{final} = \prod_{l=1}^{L} a_l = \prod_{l=1}^{L}[a_l^1, a_l^2, ..., a_l^N], \quad (17)$$

where $N$ is the number of attention heads in each attention layer. Then we introduce a selection hyperparameter $\alpha$ to select the patch features $h_{local}$ with the highest attention weights. At the same time, we introduce the maximum value of patch features in the patch dimension to fine-tune the important local information selected according to the attention mechanism. We first use a global max pooling to obtain the maximum feature $h_{max}$, then we expand it to the same dimension of $h_{local}$ and get the final local features captured as Eq. 18 shows.

$$h_{local}'^l = h_{local} + Softmax(h_{local}(h_{max})^T)h_{max}, \quad (18)$$

where $h_{local}'^l$ represents the final local feature of the $l$-th stage. Finally, we use a linear transformation to keep the final local feature of each stage as the same dimension and stack them as the local representation of the sticker, which is set as $X_{local}$. Furthermore, with the global representation $X_{global}$ we obtained before, we concatenate the $X_{local}$ to $X_{global}$ as the final visual representation.

### 3.5 Prediction Module

With the given visual feature and text feature, we first used two linear transformations to project them into the same dimension. Then we concatenate the textual feature to the visual feature and add a position embedding to it. Inspired by agent attention [9], we design

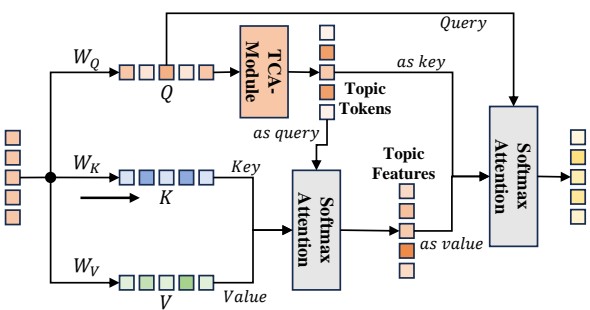

**Figure 4: Framework of the proposed TG-Attention.**

a topic-guided attention to better capture features from stickers with the same topic ID, as shown in Fig. 4. After obtaining $Q, K, V$ with linear transformation, we utilize a TCA-Module to obtain a Topic Token from the $Q$ and then feed it as the query and $K, V$ to a Softmax Attention to obtain the Topic Features. Furthermore, we use the $Q$ as the query, Topic Tokens as the key, and Topic Features as the value to conduct a Softmax Attention. In this way, we can better fuse the sticker representation and context features from the stickers with the same topic ID. Finally, we utilize an MLP and a linear prediction head to get the emotion recognition results.

## 4 EXPERIMENTATION

This section systematically evaluates our proposed TGCA-PVT for sticker emotion recognition and also extends to image-based emotion recognition.

### 4.1 Experimental Settings

**Datasets.** We evaluate our TGCA-PVT model on the benchmark sticker-based emotion recognition datasets SER30K [17] [1]. Furthermore, to demonstrate the effectiveness of our TGCA-PVT model in the field of image emotion recognition, we conduct experiments on the image-based emotion recognition dataset FI [40] [2]. Both of these two datasets are visual sentiment analysis datasets, but different from the FI dataset with only images, the SER30K dataset is a sticker dataset in which all stickers have the corresponding theme and some images have corresponding text modal information.

**SER30K** dataset contains 30,739 stickers including 1,887 stikcer themes collected from the sticker image website [3]. Each sticker is annotated with the emotion label by three annotators and belongs to a common theme. Stickers within the same theme have similar subject characters. There are 7 categories of sentiment labels (*i.e. Anger, Disgust, Fear, Happiness, Neutral, Sadness, and Surprise*, and 5,886 stickers inside are annotated with text information. More details about the samples contained in different emotion labels are provided in Table 1. Conducting emotion recognition on this dataset can help better investigate the effect of stickers on users' emotions in online chatting on social media.

**FI dataset** is a widely used large-scale visual sentiment analysis dataset obtained on two popular social platforms, Flicker and

Instagram. It contains 23,308 images with 8 sentiment categories (*i.e. Amusement, Anger, Awe, Contentment, Disgust, Excitement, Fear, and Sadness*, and more than 1,000 samples are contained by each category. There's no text information or themes about the images. Since the images have no theme, we use the file name to generate the topic id of the image.

**Table 1: Detailes of datasets used.**

| SER30K | Samples | Samples with text | FI Dataset | Samples |
|---|---|---|---|---|
| Anger | 2,750 | 439 | Amusement | 4,942 |
| Disgust | 211 | 17 | Anger | 1,266 |
| Fear | 826 | 58 | Awe | 3,151 |
| Happiness | 11,255 | 1,965 | Contentment | 5,374 |
| Neutral | 10,815 | 2,832 | Disgust | 1,658 |
| Sadness | 3,359 | 346 | Excitement | 2,963 |
| Surprise | 1,523 | 229 | Fear | 1,032 |
| - | - | - | Sadness | 2,922 |
| Total | 30739 | 5886 | Total | 23,308 |

**Implementation Details.** Using the same setup as the Ref [17], we randomly divided the SER30K data set into training, validation, and test sets at a ratio of 7:1:2. We implemented our approach based on the Pytorch framework [19]. All of the experiments in the work are conducted on NVIDIA GTX 4090. For textual input, the max sequence of features obtained by the pre-trained Bert model is set as 30, and the feature dimension is 768. For the pre-trained PVT encoder for visual features, we adopt the PVT-small [31] pre-trained on the ImageNet1k [4]. The proposed TGCA-PVT is optimized using the SGD algorithm with a learning rate of $10e^{-4}$. The size of the sticker input to the model is 448x448. We also set the batch size as 16 and the epoch numbers as 50.

### 4.2 Baselines

To evaluate the proposed method for sticker emotion recognition on the SER30K dataset, we first compare our proposed model with various baselines in image emotion analysis without text information, ranging from classical non-numerical methods such as SVM and RF to state-of-the-art deep neural models including AlexNet [15], VGG [48], ResNet [10] and ViT [6]. Additionally, we investigate advanced models incorporating attention mechanisms to capture emotional information from sticker regions effectively. Specifically, we consider WSCNet [35], PDANet [45], and LORA-V [17], which can utilize attention mechanisms to enhance emotional feature extraction from sticker images, demonstrating their potential for improving sticker emotion recognition performance. Then for stickers with text information, we compare our proposed method with some multimodal emotion recognition methods including WSCNet-T and PDANet-T, which use WSCNet and PDANet to capture the sticker information and then use the same way as LORA to encode the text feature and fuse vision feature and text feature. We also compare multimodal fusion methods including LORA [17], TFN [41], and MCB [8]. By comparing these methods, our proposed method's performance can be effectively verified.

For the FI dataset, we introduce vision sentiment analysis models including Sentibank based on psychological theories and web

**Table 2: Performance of the recently proposed emotion recognition methods in the SER30K Dataset. The precision of each emotion category as well as the overall classification accuracy are reported. All values in the table are in percentages.**

| Modality | Model | Precision on each emotion category | | | | | | | Accuracy |
|---|---|---|---|---|---|---|---|---|---|
| | | Anger | Disgust | Fear | Happiness | Neutral | Sadness | Surprise | |
| Image | SVM | 25.66 | 22.22 | 28.96 | 63.48 | 67.18 | 34.34 | 23.42 | 51.02 |
| | RF | 74.07 | 100.0 | 100.0 | 52.52 | 50.84 | 75.00 | 75.00 | 52.21 |
| | AlexNet | 20.00 | 00.00 | 00.00 | 53.33 | 49.67 | 33.13 | 10.52 | 50.87 |
| | VGG | 37.40 | 00.00 | 41.02 | 73.55 | 60.76 | 51.52 | 35.14 | 62.57 |
| | ViT | 52.72 | 32.00 | 53.70 | 74.68 | 62.92 | 56.55 | 40.31 | 64.94 |
| | ResNet | 50.30 | 26.66 | 57.01 | 76.63 | 65.94 | 64.69 | 48.33 | 67.76 |
| | PDANet | 58.10 | 26.66 | 61.68 | 79.60 | 64.76 | 63.50 | 47.10 | 68.68 |
| | WSCNet | 58.77 | 0.00 | 74.62 | 79.49 | 63.50 | 65.96 | 49.53 | 68.98 |
| | LORA-V | 54.71 | 50.00 | 64.15 | 78.04 | 67.03 | 66.25 | 44.68 | 69.22 |
| | TGCA-PVT-V | 57.17 | 57.14 | 68.42 | 76.92 | 68.96 | 65.48 | 52.40 | **70.23** |
| Image+Text | SVM | 22.61 | 9.33 | 12.36 | 65.36 | 61.99 | 33.05 | 20.02 | 45.05 |
| | RF | 90.00 | 100.0 | 100.0 | 52.81 | 51.27 | 82.22 | 66.66 | 52.55 |
| | TFN | 00.00 | 00.00 | 00.00 | 58.13 | 51.74 | 32.98 | 00.00 | 54.19 |
| | MCB | 35.16 | 18.18 | 50.70 | 64.64 | 58.66 | 49.11 | 32.93 | 58.18 |
| | PDANet | 60.09 | 19.23 | 59.29 | 80.57 | 65.02 | 61.08 | 48.98 | 68.93 |
| | WSCNet | 56.64 | 36.84 | 60.00 | 77.85 | 66.72 | 67.04 | 49.18 | 69.45 |
| | LORA | 59.18 | 50.00 | 63.09 | 75.83 | 67.78 | 67.60 | 54.14 | 70.51 |
| | TGCA-PVT | 65.67 | 35.73 | 66.09 | 79.57 | 69.39 | 63.62 | 53.04 | **71.63** |

mining [1], PAEF which extracts principles-of-art-based emotion features [44], DeepSentibank that improves the Sentibank with deep CNNs [3], deep CNNs models for image classification including Fine-tuned AlexNet [15], MldrNet that combines deep representations of different levels [22], Fine-tuned VGG16 [26], label distribution learning proposed by Yang et al. [37], Fine-tuned ResNet50 [10], the sentiment constraints and the hierarchical relation of emotion labels proposed by Yang et al. [36], MAP proposed by He et al. that performs pyramidal segmentation and pooling for visual sentiment analysis [11], WSCNet [35], a deep CNN model to extract and integrate the content information from the high layers and style information from the low layers, which is proposed by Zhang et al. [43], SOLVER that constructs Emotion Graph based on semantic concepts and visual features for visual emotion analysis [32], stimuli-aware visual emotion analysis method proposed by Yang et al. [34], MAM that incorporates different visual concepts for emotion analysis [42] and LORA-V [17]. Accordingly, we use the proposed single-modal model TGCA-PVT-V for comparison.

## 4.3 Experimental Results

Experimental results on the SER30K dataset are depicted in Table 2. There are two tasks, the first one is to use the image modality to conduct emotion recognition and the second one is to use both the image and text modalities. We provide the average accuracy of all categories and the precision of each emotion category as the same as the Ref. [17]. We can see that our proposed TGCA-PVT achieves the best average accuracy. Although some methods have higher precision on some specific categories, their overall accuracy is not as good as our designed method. On the one hand, this situation is because there is a phenomenon of class imbalance in the data set,

and on the other hand, the model is prone to emotional confusion, which leads to high precision when the accuracy is not high.

Specifically, for single-modal sticker emotion recognition, we remove the text feature encoder in the TGCA-PVT and keep the other modules the same, which is called TGCA-PVT-V. Compared to traditional machine learning methods, most of the deep learning methods have an obvious advantage in accuracy. Especially the methods that utilize spatial attention mechanisms to capture the regional emotion features. In contrast, other models do not capture the emotional features in emoticons well.

For multimodal sticker emotion recognition, text features are additional inputs. The textual information in emoticons can provide sufficient emotional features. However, the category imbalance is also more severe due to the lower number of emoticons with textual features in the SER30K dataset. As a result, the simpler model instead decreases in recognition accuracy, while the effective methods such as PDANet, WSCNet, LORA, and our proposed TGCA-PVT can achieve better accuracy. We can see that introducing textual features and multi-modal fusion can improve them by 0.25%, 0.47%, 1.29%, and 1.38% respectively.

We also compare our proposed method with the state-of-the-art image emotion recognition methods as illustrated in Table 3. We can see that our proposed TGCA-PVT-V utilizing the topic-guided context-aware module and the frequency attention module achieves the best accuracy. Compared to the LORA-V which also utilizes the vision transformer, our method also has an improvement of 0.56%. The reason such enhancement is not as pronounced as on the SER30K dataset is that on a traditional image emotion recognition dataset such as FI, there is no segmentation of image themes, and thus the theme guidance module only provides the ability to mine

local features in terms of image enhancement. Such experimental results also demonstrate the robustness of our proposed method in analyzing the emotions contained in pictures.

**Table 3: Average accuracy of each sentiment analysis model on the FI [40] dataset.**

| Method | Acc(%) |
|---|---|
| Sentibank [1] | 49.23 |
| PAEF [44] | 46.13 |
| DeepSentibank [3] | 51.54 |
| Fine-tuned AlexNet [15] | 59.85 |
| MldrNet [22] | 65.23 |
| Fine-tuned VGG16 [26] | 65.52 |
| Yang et al. [37] | 67.48 |
| Fine-tuned ResNet50 [10] | 67.53 |
| Yang et al. [36] | 67.64 |
| He et al. [11] | 68.13 |
| WSCNet [35] | 70.07 |
| Zhang et al. [43] | 71.77 |
| SOLVER [32] | 72.33 |
| Yang et al. [34] | 72.42 |
| MAM [42] | 71.44 |
| LORA-V [17] | 72.49 |
| TGCA-PVT | 73.05 |

## 5 ANALYSIS AND DISCUSSION

We also conduct ablation studies and hyper-parameter analysis to better verify the effect of our proposed method on the SER30K dataset with both image and text. Furthermore, we show some of the attention map visualization results to help understand the effect of our model on sticker emotion recognition.

### 5.1 Ablation Study

As illustrated in Section. 3, we design several modules to better improve the performance of the backbone network for sticker emotion recognition. Specifically, we conduct extensive ablation experiments to evaluate the validity of the modules we designed. The "Base" model indicates the original backbone network that is PVT-small. Then we remove our designed components from the TGCA-PVT one by one for comparison, the compared results are shown in Table. 4. We can see that compared to the backbone network PVT-small, our proposed method brings an obvious improvement. At the same time, when removing the TGCA-module or LERA-Module, the accuracy has a relatively large drop, which means these two modules greatly improve the model's performance for sticker emotion recognition. In addition, when we remove TG-Attention, we can observe that the accuracy of the model decreases by 0.44%, validating the effectiveness of the module. From the results of these ablation experiments, our designed Topic ID-based TGCA-Module with TG-Attention can effectively mine the direct contextual features of emojis under the same topic, to utilize the topics of emojis to achieve more accurate emotion recognition.

**Table 4: Ablation experiments of TGCA-PVT. We remove each module designed one by one and then report the emotion recognition accuracy. "FLA" represents the FLA-Module, "TGCA" represents the TGCA-Module, "LERA" represents the LERA-Module, and "TG-A" represents the TG-Attention in the prediction module.**

| Methods | FLA | TGCA | LERA | TG-A | Acc.(%) |
|---|---|---|---|---|---|
| Base | | | | | 68.35 |
| | | √ | √ | √ | 71.49 |
| | √ | | √ | √ | 70.97 |
| | √ | √ | | √ | 70.93 |
| | √ | √ | √ | | 71.19 |
| TGCA-PVT | √ | √ | √ | √ | 71.63 |

We further target the designed LERA module, which we analyze by removing the modules one by one from the stage of the PVT architecture. The details and experimental results are depicted in Table 5. Specifically, we started by keeping the LERA-Module for the last stage only and then added it stage by stage. We can observe that the effectiveness of the model is gradually improved as more and more stages of the LERA-Module are used. This means that by utilizing the pyramid architecture, local features at different scales can be fused more effectively, thus allowing the model to extract the rich regional information in the stickers more efficiently, and ultimately achieve accurate sticker emotion recognition.

**Table 5: Ablation experiments of LERA-Module. "Stage" indicates a stage that uses the LERA-Module.**

| Stage | 4 | 4+3 | 4+3+2 | 4+3+2+1 |
|---|---|---|---|---|
| Acc.(%) | 70.66 | 70.90 | 71.11 | **71.61** |
| F1(%) | 69.91 | 70.18 | 70.22 | **70.93** |

### 5.2 Hyper-parameter Analysis

The only hyper-parameter used in our proposed method is the selection hyper-parameter $\alpha$ used in LERA-Module. The value of $\alpha$ affects the model's ability to extract localized features. When $\alpha$ is small, we only select fewer attention weights to compute local features, and the model will only focus on image regions with larger weights, while when $\alpha$ is too large, regional features that are not related to the emotional features of the image will also be extracted by the model, which introduces redundant information, and brings about performance degradation. Therefore, it is particularly important to consider the value of $\alpha$. We selected 1 to 10 as the values of $\alpha$ for our experiments, and a comparison of the experimental results is shown in Table 6. We can observe that when $\alpha$ is 8, our proposed TGCA-PVT achieves the best accuracy and F1 score. This means that when the value of $\alpha$ is taken as 8, the model can effectively capture the features of the regions in the sticker that contain emotional information. Meanwhile, since the input image size is 448x448 and the feature map size outputted by the PVT model in the last stage is 14x14, the size of the extracted local features is 64

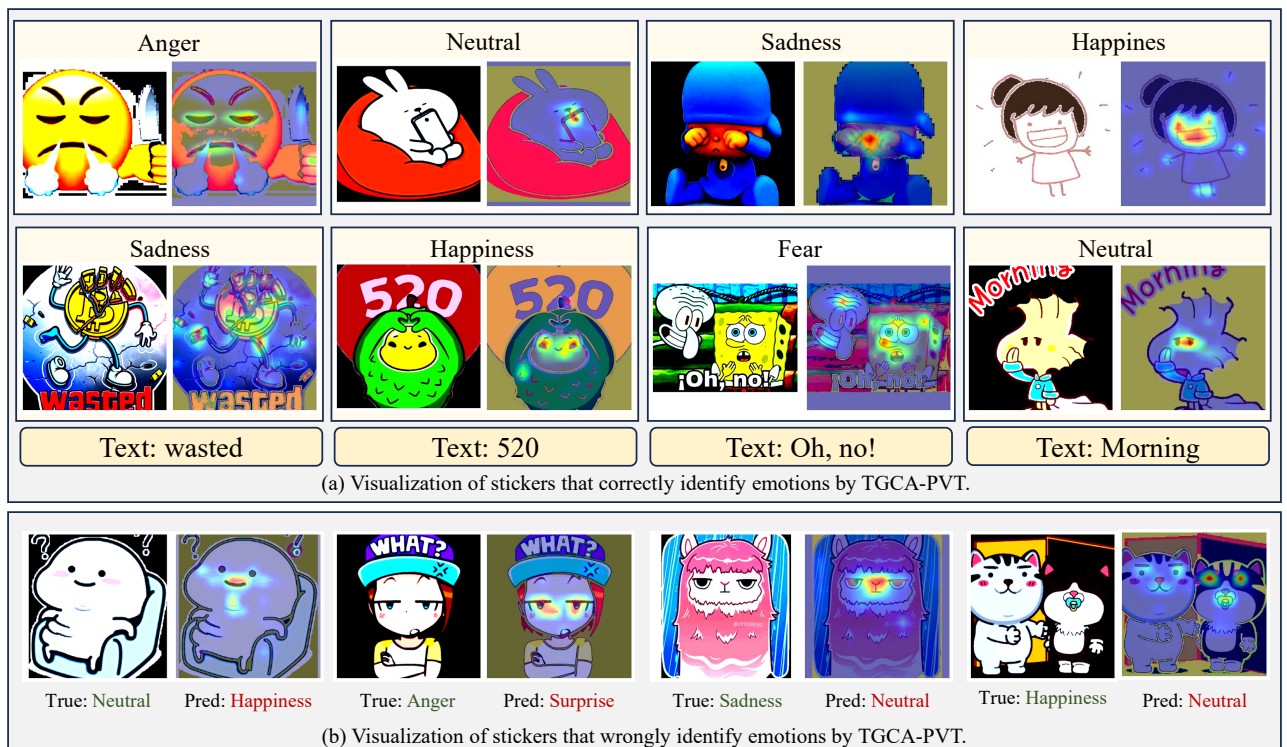

(a) Visualization of stickers that correctly identify emotions by TGCA-PVT.

(b) Visualization of stickers that wrongly identify emotions by TGCA-PVT.

**Figure 5: The visualization results of the proposed TGCA-PVT on the SER30K dataset with attention weights. For each sticker, we show the original image on the left and the attention weight visualization results on the right.**

when $\alpha$ is 8, which can contain close to 1/3 of the region of the feature map.

**Table 6: Performance of TGCA-PVT with different values of $\alpha$ in LERA-Module.**

| $\alpha$ | 1 | 2 | 3 | 4 | 5 |
|---|---|---|---|---|---|
| Acc.(%) | 71.18 | 70.86 | 71.26 | 71.15 | 70.87 |
| F1(%) | 70.65 | 70.31 | 70.59 | 70.36 | 70.14 |
| $\alpha$ | 6 | 7 | 8 | 9 | 10 |
| Acc.(%) | 70.85 | 70.95 | **71.61** | 71.24 | 71.51 |
| F1(%) | 70.31 | 70.13 | **70.93** | 70.41 | 70.89 |

### 5.3 Qualitative Results

To better show the performance of our proposed TCGA-PVT, we conduct some visualization results with the attention weights obtained from the last stage in the visual encoder. Specifically, we average the last attention weights of the last stage and then blend them into the original image to conduct visualization, as depicted in Fig. 5. We can observe that the attention mechanism in the visual coder pays more attention to the key parts of the sticker subject's eyes, mouth, and other emotional flow, and at the same time, when there are two subjects, the attention mechanism also embodies the attention on both at the same time, which makes the model achieve

a more accurate emotion recognition result. However, excessive attention to these regional features can also cause the model to misrecognize, as in the first three misclassification examples in (b), although the model pays attention to the key regions, the neglect of other features leads to incorrect recognition results. In the last misclassification example, the model is not able to distinguish the primary and secondary relationships between multiple subjects well, thus focusing more on the secondary subjects, which eventually leads to misclassification. This also analyzes the shortcomings of the existing methods and provides us with ideas for further work.

### 6 CONCLUSION

In this paper, we designed a novel topic-guided context aware to conduct better sticker emotion recognition. We introduce the concept of topic ID to help the model learn the common subject features of the same topic stickers and also design several modules to improve the performance based on the pre-train vision encoder PVT-small and text encoder Bert. Extensive experimental results on the large-scale sticker emotion dataset SER30K and the image emotion recognition FI dataset verify the effect of our proposed TGCA-PVT. We also find that making the model recognize the multi-subject nature of stickers and distinguish the relationship between different subjects is the key to further improving the emotion recognition results of stickers. We hope that this work will contribute to advancing the understanding of emotion recognition in online chat as well as emotion understanding.

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
