# OpenReview forum: "TGCA-PVT: Topic-Guided Context-Aware Pyramid Vision Transformer for Sticker Emotion Recognition"
_acmmm.org/ACMMM/2024/Conference — MM2024 Poster_

### Official Review · Reviewer_C5CE · 2024-05-05

**Rating:** 2
**Confidence:** 3

**Summary:**

Unlike traditional image emotion recognition, sticker emotion recognition requires considering both global and local features, as well as additional modalities like text. To address this, a topic ID-guided transformer method is introduced, facilitating a nuanced analysis of stickers. Each sticker is assigned a topic ID, allowing stickers with the same topic to be grouped for context analysis. Experiments on SER30K and FI datasets validate the effectiveness of the method.

**Strengths:**

1. This paper presents the concept of Topic ID and suggests a TGCA-Module and TG-Attention based on Topic ID to extract the shared subject information among emoticons with the same topic, alongside enhancing local features through image transformation.

2. This paper devises an FLA module to improve the capture of frequency domain information for extracting sticker object features, along with an LERA module to enhance the model's capability in extracting local details of stickers for improved emotion recognition.

3. Extensive experiments are conducted on SER30K and FI datasets.

**Limitations:**

1. My first confusion is why research sticker emotion recognition, what is its significance? Compared to research on image emotion recognition itself, what are the difficulties and challenges?

2. How to determine the number of topic IDs? In real-world scenarios, it's not feasible to estimate the number of topics in advance.

3. In experiments, comparisons should be made with existing image emotion recognition methods on the SER30K dataset.

4. If the sticker input lacks textual information, what should be done?

**Suitability:**

2

---

### Official Review · Reviewer_w5V7 · 2024-05-20

**Rating:** 4
**Confidence:** 4

**Summary:**

This paper proposes a TGCA-PVT model to incorporate both global and local features in sticker emotion recognition, which consists of a topic-guided context-aware module and a topic-guided attention mechanism to extract topic context features, along with a frequency linear attention module to better capture the object information.

**Strengths:**

1 The overall writing of this paper is clear, the method level is clear and easy to understand.

2 This paper introduces the concept of Topic ID and proposes a TGCA-Module and a TG-Attention to mine the subject information.

3 This paper designs a frequency linear attention module to better capture the object information and a LERAModule to extract local details of stickers.

4 This paper conducts a relatively comprehensive experiments.

**Limitations:**

1 For motivation, why is it important to consider both local and global information in SER tasks? It is not obvious from the examples given in Figure1, and it seems that the study of global and local information in the image is not very meaningful

2 For approach:

(1) Is the TopicID in the method already annotated in the data set, or is it built by yourself?

(2) What is the Imaginary Part of Figure 3? There seems to be no introduction in the body of section 3.2.

(3) Is the whole model improved on the basis of PVT? But why does the experimental results in Table 2 seem not to be compared with PVT?

(4) Section 3.5 seems to be missing a description of the loss function, which is essential for the repetition method.

(5) According to the statistics in Table1, the number of text and the number of image are not equal. Then how does the model deal with the problem of missing text?

3 For experiments:

(1) It seems that the classification task is not complete enough to only evaluate Acc, F1 value is equally important as Acc, and according to the data statistics in Table 1, SER has a serious category imbalance problem, so W-F1 value is very important.

(2) I am curious about the performance of existing multimodal large language models, such as LLaVA and GPT4-V on SER tasks, and suggest comparing the results of the two large models to enrich the experimental content and make the results more convincing.

(3) The SOTA of each emotion category in Table 2 should also be marked in bold to facilitate the analysis of experimental performance.

**Suitability:**

2

---

### Official Review · Reviewer_nAjB · 2024-05-24

**Rating:** 2
**Confidence:** 3

**Summary:**

For emotion recognition of stickers, this paper proposes the TGCA-PVT which integrates global and local features and textual information. TGCA-PVT extracts global information from stickers with the same topic and integrates a frequency linear attention module to better capture the object information of the stickers and a locally enhanced re-attention mechanism for improved local feature extraction.

**Strengths:**

-	This paper deals with the interesting Sticker Emotion Recognition task.
-	The figures are subtly designed, which is beneficial for readers in understanding the paper.

**Limitations:**

-	As the most relevant work to the manuscript, it is necessary to discuss the differences and advantages of the proposed method compared to LORA[1]. While Section 3.4 briefly mentions that the LERA module is inspired by LORA, the rest of the text, including the related work, lacks further comparison. Additionally, the novelty of the proposed method seems limited, further elaboration is needed to clarify the improvements of this work.
-	I noticed discrepancies between the results of LORA reported in this paper and those in the original paper[1]. For example, in Table 2, LORA achieves 70.51 on Image+Text, while [1] reports 70.73. Similarly, [1] reports the result of LORA as 73.32 in Table 3, which performs better than TGCA-PVT. Only the results of LORA among the compared methods differ from [1]. It would be necessary to investigate whether different settings are used or if there are any explanations for this phenomenon.
-	As the FI dataset lacks the topic structure and the content and style of the images under each emotion category vary widely, using the emotion category as theme ID may not be entirely appropriate. I am intrigued by the potential insights TGCA-PVT can offer on the FI dataset. It would be beneficial to conduct additional experiments or analyses to explore what information TGCA-PVT can capture on common image emotion datasets such as FI.

[1] Liu S, Zhang X, Yang J. SER30K: A large-scale dataset for sticker emotion recognition, In ACM MM. 2022.

**Suitability:**

3

---

### Official Review · Reviewer_PnPh · 2024-06-02

**Rating:** 5
**Confidence:** 4

**Summary:**

proposed a method, TGCA-PVT, for sticker emotion recognition.

**Strengths:**

1. integrated a frequency linear attention module to leverage frequency domain information to better capture the object information of the stickers.
2. a locally enhanced re-attention mechanism for improved local feature extraction

**Limitations:**

1. Similar to the multimodal classification task， only the acc metric is not enough, such as recall, presion, map, etc. It's best to visualize the sentiment distribution and see how the classification behaves.
2. Complementing the latest models.
3. The SER30K dataset scenario looks simple, and the reason for the generally low acc can be briefly analyzed.
4. It is necessary to analyze the time complexity.

**Suitability:**

3

---

### Meta-Review · Area_Chair_jXRu · 2024-06-30

**Recommendation:** Accept (Poster)
**Confidence:** 3

**Metareview:**

The submitted paper addresses the task of Sticker Emotion Recognition (SER). The work is well-structured, well written, and introduces several novel concepts to enhance the accuracy and efficacy of emotion recognition from sticker images. The strengths and limitations as highlighted by the reviewers are as follows:

Strengths:

The paper addresses the interesting and relatively unexplored task of Sticker Emotion Recognition, contributing to the growing field of multimodal emotion recognition.
The paper is well-written, with a clear and logical flow. The figures are well-designed, aiding in the understanding of the proposed methods and results.
The paper designs a Frequency Linear Attention (FLA) module to capture object information and a Local Enhanced Residual Attention (LERA) module to extract local details, both of which are innovative and technically sound.
Extensive experiments are conducted on the SER30K and FI datasets, demonstrating the efficacy of the proposed methods. The experimental results are thorough and provide a strong validation of the model's performance.

Limitations:

As the FI dataset lacks a topic structure and has widely varying content and style under each emotion category, using the emotion category as the theme ID may not be entirely appropriate. Additional experiments or analyses on what TGCA-PVT can capture on common image emotion datasets such as FI would be helpful.
The classification task should not only evaluate accuracy (Acc) but also include F1 score, especially considering the category imbalance problem in the SER dataset.
Comparisons with existing multimodal large language models like LLaVA and GPT-4V on SER tasks would enrich the experimental content and make the results more convincing.
The significance of researching sticker emotion recognition as opposed to general image emotion recognition is not clearly articulated. The specific challenges and difficulties unique to sticker emotion recognition should be better explained.

The paper makes a substantial contribution to the field of emotion recognition through its novel task, comprehensive dataset, and innovative methodologies. While there are several areas for improvement and clarification, these do not detract from the overall quality and potential impact of the work.

Given the strengths of the paper and the constructive feedback provided by the reviewers, I recommend that this paper be accepted for presentation at the conference. The authors are encouraged to address the reviewers’ feedback in their final revision to further enhance the paper's contributions and impact.